# The Other Side of Malnutrition in Inflammatory Bowel Disease (IBD): Non-Alcoholic Fatty Liver Disease

**DOI:** 10.3390/nu13082772

**Published:** 2021-08-13

**Authors:** Giulia Gibiino, Alessandro Sartini, Stefano Gitto, Cecilia Binda, Monica Sbrancia, Chiara Coluccio, Vittorio Sambri, Carlo Fabbri

**Affiliations:** 1Gastroenterology and Digestive Endoscopy Unit, Ospedale Morgagni-Pierantoni, AUSL Romagna, 47121 Forlì, Italy; ale.sartini@gmail.com (A.S.); cecilia.binda@gmail.com (C.B.); monica.sbrancia@auslromagna.it (M.S.); chiara.coluccio@auslromagna.it (C.C.); carlo.fabbri@auslromagna.it (C.F.); 2Gastroenterology and Digestive Endoscopy Unit, Ospedale M.Bufalini, AUSL Romagna, 47521 Cesena, Italy; 3Department of Experimental and Clinical Medicine, University of Florence, 50100 Florence, Italy; stefano.gitto@unifi.it; 4Unit of Microbiology, The Great Romagna Hub Laboratory, 47522 Pievesestina, Italy; vittorio.sambri@auslromagna.it; 5Unit of Microbiology, DIMES, University of Bologna, 40125 Bologna, Italy

**Keywords:** non-alcoholic steatohepatitis, leaky gut, metabolic syndrome

## Abstract

Steatohepatitis and hepatobiliary manifestations constitute some of the most common extra-intestinal manifestations of Inflammatory Bowel Disease (IBD). On the other hand, non-alcoholic fatty liver disease (NAFLD) affects around 25% of the world’s population and is attracting ever more attention in liver transplant programs. To outline the specific pathways linking these two conditions is a pressing task for 21st-century researchers. We are accustomed to expecting the occurrence of fatty liver disease in obese people, but current evidence suggests that there are several different pathways also occurring in underweight patients. Genetic factors, inflammatory signals and microbiota are key players that could help in understanding the entire pathogenesis of NAFLD, with the aim of defining the multiple expressions of malnutrition. In the current review, we summarize the most recent literature regarding the epidemiology, pathogenesis and future directions for the management of NAFLD in patients affected by IBD.

## 1. Introduction

Today, non-alcoholic fatty liver disease (NAFLD) represents one of the most common causes of liver disorder, affecting 17–46% of adults in developed countries. NAFLD is characterized by the presence of steatosis in a percentage of hepatocytes greater that 5% [1]. Notably, its diagnosis requires the exclusion of secondary causes of liver disorder and significant daily alcohol intake (≥30 g for males and ≥20 g for females) [2]. In general, NAFLD should be considered the hepatic expression of metabolic syndrome (MetS), with a high prevalence that follows obesity and other metabolic disorders [3]. Consequently, obesity, insulin resistance, diabetes mellitus, hypertension, and hypertriglyceridemia are the main risk factors for NAFLD [4]. Remarkably, NAFLD can potentially progress from “simple” steatosis to non-alcoholic steatohepatitis (NASH), liver fibrosis (LF), cirrhosis and hepatocellular carcinoma (HCC) [3].

Crohn’s disease (CD) and ulcerative colitis (UC) represent the two most important forms of inflammatory bowel disease (IBD). They are both immune-mediated, chronic-relapsing inflammatory illnesses that definitely affect the gut [5,6,7]. Notably, both CD and UC can be coupled with extra-intestinal manifestations including the hepato-biliary diseases (5–50% of cases) [8,9]. Among them, NAFLD plays a major role, accounting for 40% of the hepatic alterations diagnosed in patients with IBD [10,11].

IBD patients are at increased risk of developing NAFLD, and the prevalence of NAFLD in IBD subjects ranges from 8% to 59%, varying according to the diagnostic criteria used [12,13,14,15,16,17]. The prevalence of non-alcoholic fatty liver disease has been reported to be between 1.5% and 55% in ulcerative colitis and 1.5% and 39.5% in Crohn’s disease [18]. However, it should be remembered that primary sclerosing cholangitis (PSC) is also part of the biliopancreatic spectrum of IBD patients. In this population, the presence of PSC is a protective factor against the development of NAFLD despite the metabolic profile [14].

Both gut inflammation and metabolic factors are concurrent with the pathogenesis of IBD-associated NAFLD. As mentioned previously, classic metabolic risk factors play a major role not only in patients with NAFLD alone, but also in subjects with both NAFLD and IBD when considering the same risk factors as the general population [19].

Regarding prognosis, it is conceivable that each disease could negatively influence the outcome of the others. In fact, 5% of IBD patients develop a severe liver disease [20]. On the other hand, NAFLD, especially in its progressive form (NASH), might make clinical management of IBD particularly difficult [21]. Notably, hospitalized IBD subjects with concomitant liver disease show a 2-fold higher mortality than patients with IBD alone [22].

The main underlying causes and predisposing risk factors to NAFLD among IBD patients remain only weakly examined. It is necessary to more thoroughly understand the pathophysiology of NAFLD development and disease progression in IBD patients, as well as the impact of NAFLD itself on bowel disease. In this complex multidisciplinary context, systemic and local inflammation and the gut–liver axis might be two key factors. Indeed, the main clinical and scientific point of discussion is related to the possible impact of one disorder on the other.

We have on the table the following unsolved questions: What is the risk profile of IBD patients regarding the liver disease? What the role of nutritional status? Do patients with concomitant liver and bowel disease show a different natural history? Do they display a worse prognosis? Is the disease control and management different in comparison to subjects with only one of the two disorders? Do the therapeutic options diverge?

The aim of the present review article is to thoroughly analyze these open issues with a critical and multidisciplinary approach.

## 2. Methods

We selected articles discussing the association of NAFLD and IBD, paying specific attention to metabolic profile. We developed a non-systematic review article using the following electronic sources: PubMed, EMBASE, Google Scholar, Ovid, MEDLINE, Scopus, the Cochrane controlled trials register, and Web of Science. We used the following search terms singly and in combination: “NAFLD”, “NASH”, “inflammatory bowel disease”, “ulcerative colitis (UC)”, Crohn’s disease (CD)”, “gut–liver axis”, “dysbiosis”, “inflammatory bowel disease AND microbiota modulation”, “NAFLD AND metabolic profile in IBD” “NAFLD AND IBD AND obesity”, “NAFLD AND IBD AND malnutrition”. We examined all the articles reporting data related to humans (inclusion criterion), while excluding works with no full text available, works that were not in the English language, book chapters and abstracts, and articles published before 1990 (exclusion criteria). Finally, we evaluated supplementary references among the articles evaluated in the first search round.

## 3. NAFLD in IBD Patients with “Metabolic Syndrome Profile”

The prevalence of obesity in IBD patients is increasing, presenting numbers that are worrying, just like in the general population [23]. From 15% to 40% of adults with IBD are obese, and a further 20–40% are overweight [24,25]. This could be a direct effect of the aggressive medical therapy currently available, and represents a warning regarding the future management of obesity in younger populations. Persistent remission of disease is the main factor linked to weight gain, more so than medications, which were not significantly responsible for this metabolic profile [26]. McGowan et al. identified the improvement of therapies and nutritional status as being a cause of increased BMI among patients affected by quiescent Crohn’s disease, with there being a direct relationship between prolonged remission, higher risk of MetS, and development of NAFLD [21].

According to some recent studies, these patients often exhibit established features of metabolic syndrome, which is conventionally considered the main factor leading to NAFLD in overweight people [12,13,15].

Dysbiosis and altered gut signaling induced by IBD, acting through hormones, satiety-related peptides, and bile acids, could be responsible for the onset of obesity and dysmetabolism. Smoking cessation and the use of corticosteroids in this setting can also contribute to weight gain [27,28]. Just as in the non-IBD population, the establishment of NAFLD is conventionally explained by the “multiple hit” hypothesis, which considers multiple insults acting together on genetically predisposed subjects. This theory includes aspects such as insulin resistance, hormones secreted from the adipose tissue, nutritional factors, gut microbiota and genetic and epigenetic factors [29]. From this point of view, a disruption of the intestinal barrier could be the first point to trigger the progression of hepatic damage. The concept of the gut–liver axis was introduced as a way of better understanding the complexity and continuously emerging elements in this mutual relationship [30]. Further evidence is also emerging regarding the main signaling involved in this pathogenesis, i.e., chemokines [31].

It has long been known that patients with IBD are vulnerable to vitamin deficiencies [32], and most recent research has emphasized some possible immunomodulatory functions linked to the development of liver disease. This is related to vitamin D and its receptor, for example, and may also involve the overweight patient population. Although recent studies are still rather heterogeneous, interesting data are emerging regarding the potential role of vitamin D in NAFLD. A possible link has been proposed between vitamin D supplementation and positive effects on liver damage in overweight and obese patients [33]. Vitamin A is also involved in hepatic lipid metabolism and integrates with adipose tissue and insulin to become a potential factor involved in NAFLD [34]. Finally, folates have similarly been observed to be important factors in the development of NASH and NAFLD in obese people [35]. Despite the curious question as to the status of micronutrient values in IBD patients who develop NAFLD, to date, the main population studied in this setting has not been evaluated in these terms. However, future studies will illustrate whether this is a coincidence or if it is a future area of study that can be modulated by diet and nutritional supplementation.

Even in the presence of metabolic syndrome, innate genetic predisposition seems to have been gaining increasing importance in recent years. NAFLD has been associated with a variety of single nucleotide polymorphisms of genes involved in lipid and glucose metabolism and immunoregulation [36]. The PNPLA3 protein is a lipase critically involved in intrahepatic lipid metabolism. Mancina et al. showed for the first time, in two Italian cohorts, that PNPLA3 148M carriers with IBD have a higher risk of hepatic steatosis and higher biomarkers of liver damage [37]. Obese carriers of the PNPLA3 mutation are predisposed to HCC development, and early diagnosis could provide the identification of a subgroup with a worsened prognosis [38]. In other words, the genetic contribution to liver disease is still quite controversial and could explain the onset of the disease, even independently from the metabolic profile. Patients responding to this metabolic profile are more likely to have higher transaminase levels and NASH than patients without metabolic syndrome [15].

## 4. NAFLD Warning in Underweight and Lean Patients

It is necessary to consider that some studies support the hypothesis that IBD-related liver damage can occur even in the absence of obesity and insulin resistance [39]. Some patients affected by NAFLD and IBD show fewer metabolic risk factors than NAFLD patients [40]. This leaves many open questions regarding the contribution of other factors, and prompted our idea that there may be a metabolic profile that is at particular risk for liver damage.

Undernutrition is a common condition among patients affected by active inflammatory bowel disease, and can lead to a condition of liver dysfunction. Malnutrition with significantly low weight is generally defined by a body mass index (BMI) <18.5 kg/m^2^ [41,42]. Malnutrition in active IBD has a prevalence in the range of 25.0–69.7% among the whole IBD population, and the frequency of severe malnutrition is in the range from 1.3% to 31.6% [43]. This condition is particularly common in patients with active Crohn’s disease, but nutritional decline can also be observed in patients with active ulcerative colitis [44,45].

Understanding the link between undernutrition and liver injury is still a complex and an unexplored field. Chen et al. recently described some pathophysiological features in a human population and in experimental mouse models to understand the onset of “lean NAFLD” [46]. They studied 538 Caucasian lean patients affected by biopsy-proven NAFLD. The model proposed supported the conclusion that these patients were obesity resistant, adopting compensatory mechanisms including increases in bile acid serum levels and fibroblast growth factor (FGF) 19 activity, as well as adopting a different gut microbiota profile; these markers were also significant when comparing early versus advanced liver fibrosis. Furthermore, they investigated altered bile acid levels and the microbiome in a murine model, confirming these significant alterations.

Understanding these mechanisms in patients affected by IBD is even more complex and remains largely unexplored. Patients with severe bowel disease are often underweight, and some mechanisms of immune activation could provide an explanation for their liver status. Along the same lines, supporting the notion that increased inflammation plays a relevant role in NAFLD pathogenesis is the finding that antiTNFα drugs exert a protective effect toward severe liver steatosis. Therapy with antiTNFα has also been found to be the only independent factor positively influencing altered liver enzymes in this subgroup of patients [39]. Carr et al. made an important contribution regarding the influence of MetS on NAFLD severity in patients with IBD. However, they recognized that the majority of IBD patients in their cohort (77%) had NAFLD in the absence of MetS; this suggested that IBD patients develop NAFLD as a result of an increased inflammatory load, and not because of metabolic risk factors [15].

An important clinical aspect in recent years has been the presence of sarcopenia, now defined as an acute or chronic skeletal muscle disease [47], and a great deal of attention is being directed towards the early diagnosis of this condition in IBD patients.

The possible link between sarcopenia and NAFLD is not a new insight, considering that reduced muscle mass leads to insulin resistance and intrahepatic gluconeogenesis with the deposition of free fatty acids [48]. An interesting recent study including 488 IBD patients assessed the study population for NAFLD, defined on the basis of a CT-scan, and sarcopenia. NAFLD was present in 11.1% of the study population and the prevalence of sarcopenia was 34.9%, suggesting that it was an independent risk factor for NAFLD after taking age, sex and metabolic factors into consideration [49].

Lean IBD patients are often affected by multiple macro- and micronutrient deficiencies [50]. As mentioned above, micronutrients could be involved in metabolic syndrome, but it is reasonable to expect that they also affect liver damage in lean people.

Key NAFLD risk genes that have been recently identified could be important factors in explaining the occurrence of NAFLD in the absence of metabolic syndrome. The PNPLA3 gene was the first gene introduced in recent years as a result of the innovations provided by genome-wide association studies (GWAS) [36]. The PNPLA3 variant seems not only to be a risk factor for developing fatty liver, but, even more interestingly, has been shown to be responsible for the degree of hepatic injury [51]. In patients with NAFLD occurring in the absence of obesity, the PNPLA3 p.I148M allele is more frequent than in other NAFLD patients, and has been independently associated with NASH and liver fibrosis [52]. Its specific contribution in IBD patients has already been demonstrated [37]. One interesting factor is the link between the PLPLA3 variant and low serum retinol levels, confirming the importance of vitamin A and its implications in the progression of liver disease [34]. Among other genes that have been studied in this setting, Transmembrane 6 superfamily member 2 (TM6SF2) is another gene that promotes susceptibility to NAFLD when expressed in the variant p.E167K [53]. Genetic variant studies are extraordinarily complex and are not a part of everyday practice. However, they could be useful for disease prediction and disease surveillance, especially in underweight patients with unexpected NAFLD.

According to the population recently studied by Adams et al., underweight patients with IBD-related NAFLD tended to be younger and did not have signs of NASH or advanced liver fibrosis, suggesting that NAFLD could be an early stage of liver damage prior to the emergence of steatohepatitis and fibrosis, or even a distinct type of fatty liver disease [54].

Other long-term follow-up studies showed that lower BMI was linked to lower stages of fibrosis, but these patients were at higher risk for the development of severe liver disease compared to patients with NAFLD and a higher BMI, independent of the presence of any confounders [55]. This confirms once again that NAFLD in lean or underweight patients is not a simple benign condition and will require accurate diagnosis and staging in the future.

## 5. Beyond the Gut Barrier and Future Perspectives in Malnutrition

The intestinal microbiota has acquired a central role in recent years in intestinal and extra-intestinal disease and, according to the evidence mentioned above, it surely plays a key role in the mechanisms underlying liver damage in patients affected by IBD. Approaching the gut microbiota involves firstly the gut barrier and the local microenvironment. The barrier is constituted of several interconnected elements including microorganisms, extracellular factors, epithelial cells, the immune system, and the vascular network [30]. Our microbiota also contributes to several signaling pathways, being able to communicate with extra-intestinal sites, as a result of the gut–liver axis and the gut–brain axis. *Firmicutes* and *Bacteroidetes* are the two dominating phyla of our gut, followed by *Proteobacteria*, *Actinobacteria*, *Fusobacteria* and *Verrucomicrobia*. Archaeal, fungal and viral components are also key players in this system. These main bacteria are responsible for absorption, storage and energy balance from dietary nutrients [56].

For at least twenty years, it has been known that there is a link between intestinal permeability, endotoxemia and NAFLD [57,58]. Referring to the ‘multiple hit’ pathogenesis of NAFLD mentioned above, the gut microflora is the first player leading to intestinal fatty acid production, resulting in increased gut permeability and elevated levels of portal bacterial endotoxin. This could act as a trigger for liver inflammation, steatohepatitis, and fibrosis.

On the other hand, IBD pathogenesis has been shown to be correlated with an exacerbated immune response to commensal microbiota, leading to continuous and worsening intestinal wall inflammation [59,60]. It is therefore reasonable to expect a link between IBD and NAFLD sustained by the complex system of our microbiota.

A specific description of the microbiota mechanisms is beyond the scope of this review. However, we would like to highlight some recent evidence regarding specific pathways that could help the future clinical management and early diagnosis of these patients. The gut microbiota can induce liver inflammation by providing toll-like receptor ligands (e.g., LPS, peptidoglycan, bacterial flagella and DNA), which promote downstream signaling events, thus leading to the secretion of proinflammatory cytokines [61]. Given the evidence of immune hyperactivation in IBD, it is reasonable to expect a similar mechanism, perhaps amplified, in patients with IBD presenting with the onset of NAFLD. Interesting evidence is emerging regarding the possible activation of T cells from the gut barrier in response to extra-intestinal targets in IBD patients. The presence of leaky gut and imbalance with respect to the immune response at this level has led to interesting theories regarding the possible recruitment of aberrant T cells to extra-intestinal sites such as the liver [62]. This would also be in agreement with the now-accepted pathogenetic model for the association of sclerosing cholangitis and IBD.

Microbiota alterations according to metabolic profile are an area of great interest that still remains partly unexplored. Current evidence reports heterogeneous data regarding the composition of the microbiota in obese patients. Some studies have shown an increased proportion of Firmicutes and reduced concentrations of Bacteroidetes in obese compared to lean humans and mice, while others have reported that there are no significant alterations in microbial composition between the two groups, with some even reporting inverse findings [63,64]. Unfortunately, the heterogeneous data and the numerous limitations associated with studies performed to explore this issue do not make it possible to state whether the alterations of the gut microbiota are a secondary phenomenon in obesity or a primary cause. However, there is consistent agreement regarding the presence of reduced diversity and altered SCFA composition and inflammation signals [65]. Recent evidence suggests a specific microbiota profile in the lean NAFLD population. Chen et al. showed significant alterations at the genus level, with enrichment in lean NAFLD patients of the Clostridiales order, including Ruminococcus, Clostridium sensu stricto 1, Romboutsia, and Ruminococcaceae UCG-008. This microbic profile is significantly different when considering lean NAFLD versus lean healthy controls, with increased *Dorea* and reduced *Marvinbryantia* and *Christensellenaceae R7* being observed in the first group [46].

Fecal microbiota transplantation (FMT) has, to date, been regarded as a safe and effective treatment, and is recommend for recurrent C. difficile infection [66]. However, there are several areas of research in which future application of FMT is being explored, including in cases of IBD and NAFLD. Unfortunately, the extreme heterogeneity of the studies, as well as their limited number, does not make it possible to validate certain data in this setting, although results suggest that in the future it could certainly become a hypothetical target therapy for the modulation of both of these diseases [67].

## 6. Conclusions

Inflammatory bowel disease often leads to nutritional imbalance. Liver function and the wide spectrum of non-alcoholic liver injuries observed in these cases is an increasingly important point that is still not completely understood. On one hand, IBD patients can develop fatty liver because of the sustained remission of bowel disease, often due to weight gain and obesity, as in the general population; on the other hand, weight loss, malnutrition and chronic inflammation can promote fat deposition within the liver. The difference in terms of predictors, natural history, and the significance of NAFLD in this setting will be a challenge for gastroenterologists and nutritionists.

Disease activity, duration of IBD and prior surgery are predictors of NAFLD development. This should represent one more incentive to achieve and maintain early clinical remission. Further prospective studies would be of interest.

## Data Availability

Not applicable.

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
