# Peer review of "The Other Side of Malnutrition in Inflammatory Bowel Disease (IBD): Non-Alcoholic Fatty Liver Disease"

_nutrients, 2021, doi:10.3390/nu13082772_

Round 1

Reviewer 1 Report

I appreciate the authors for revising this review article entitled with The other side of malnutrition in inflammatory bowel disease 2 (IBD): Non-alcoholic fatty liver disease” led by Giulia Gibiino et al. However, there are still some unconnected sentences to the topic of the article. For instance, lines 112 to 121. Authors have discussed Vitamin D in IBD, and Vitamin A and folate in NAFLD. These sentences suggest IBD and NAFLD associate different malnutrition, need more evidence to suggest the common link (malnutrition) between IBD and NAFLD. Instead, the authors stated the role of vitamin D in liver damage in obese patients. Likewise, in the article, the more strong pieces of evidence need to demonstrate the common associations only in IBD and that leads to NAFLD or vice versa.

Author Response

We thank the reviewer for this revision and understand his comments regarding any unclear connections. We added an explanatory detail to say that there is a lack of evidence in this setting, however being a narrative review, it was in our interest to name the topic to draw the reader's attention to future areas of research. In addition, this paragraph was added during the previous revision on the advice of the reviewer and was not part of the original manuscript at the beginning.

Reviewer 2 Report

The manuscript titled "The other side of malnutrition in Inflammatory bowel disease (IBD): Non-alcoholic fatty liver disease" presented by Gibiino and co-workers reviews the epidemiology pathogenesis and management of NAFLD in IBD patients. This kind of review is relevant in the context of the growing incidence of this disease. The manuscript is well written, interesting to read and the references are updated. The introduction is appropriate and the recent papers have been well discussed.

Author Response

We thank the reviewer for the positive comments and we hope he will appreciate also the definitive version.

Round 2

Reviewer 1 Report

I appreciate the authors for revising the manuscript appropriately.

This manuscript is a resubmission of an earlier submission. The following is a list of the peer review reports and author responses from that submission.

Round 1

Reviewer 1 Report

I would like to congratulate the authors for writing a review on the interesting topic that recently gaining more attention -malnutrition associated Non-alcoholic fatty liver disease. The article entitled “The other side of malnutrition in Inflammatory bowel disease (IBD): Non-alcoholic fatty liver disease”., However, the authors have not discussed much regarding malnutrition associated IBD that leads to NAFLD, but discussed more other factors including inflammation, and microbiota. I would suggest the authors discuss more nutritional factors that lead to NAFLD to justify the title. There are a number of articles that demonstrated deficiencies in vitamins (A, B, and D, etc,), ketone bodies, protein deficiency are associated with NAFLD. 

Also, please correct some typo errors in the manuscript, eg: often due to weight gain and obesity.

Author Response

We thank the reviewer for his revision and comments. We added some details about most recent evidence of micronutrient deficiencies linked to NAFLD onset. We also corrected typo errors.

Reviewer 2 Report

The review manuscript entitled “The other side of malnutrition in inflammatory bowel disease 2

(IBD): Non-alcoholic fatty liver disease” led by Giulia Gibiino et al, summarize the most recent literature about epidemiology, pathogenesis and future directions for the management of NAFLD in patients with IBD. This manuscript is solely based on author’s recent manuscript in Cell Death and Disease in 2018. However controversy exist on the main objective of this review.

Major Comments:

  1. Authors should acknowledge and review the literature and present both positive and negative association of IBD with NAFLD or NAFLD with IBD (both UC and CD)
  2. Ref17 cited in this manuscript show that incidence of NAFLD in IBD patients does not deferent from general population is the key and has been omitted in the review
  3. Authors should acknowledge that Italian population has decreased incidence of obesity and metabolic syndrome including NAFLD compared to the rest of the world
  4. The methods used to the literature search suggest that they went up to 120 years of literature, but importantly missed some of the details like NAFLD patients are older with IBD and there were only 8.2% incidence of IBD in NAFLD
  5. Details of Multiple hit hypothesis of NAFLD should be included: are authors suggesting that IBD is one of the hit or the inflammatory damage in intestine is not clear
  6. Line 117-120: How IBD is connected to cardiovascular disease can be removed from the review
  7. Authors cannot connect Anorexia Nervosa with IBD. Anorexia Nervosa is an intense fear of weight gain, starvation, protein-energy malnutrition or marasmus. Patients with AN misuse laxative, diuretic or enema and it is a psychological disorder
  8. MCD diet cannot be compared for malnutrition, choline or methionine deficient diet cannot be compared with protein malnutrition, that we see in the poor countries
  9. Carr et al mention in page 4 showed that severity of NAFLD in IBD is associated with metabolic syndrome not with the severity of IBD. Further it was suggested that metabolic syndrome predict NAFLD not the severity of IBD. This finding is in contrast to the authors recent finding in Cell death and disease publication
  10. Authors should consider analyzing the increased progression of NASH in IBD and that will prove their hypothesis discussed in line 212-217
  11. Discussion of AN in line 231 is unrelated to IBD or NAFLD
  12. The epidemiology, pathogenesis and future direction for the management of NAFLD in IBD patients is missing that was initially mentioned in the abstract for review

Minor comments

Line 99-100 is not clear to this reader. What was the theory and how these factors are unclear?

Line103: Which pathogenesis IBD or NAFLD

Author Response

Dear Editor and Dear Reviewer, 

please find in attachment our reply with detailed point-by-point answers.

Round 2

Reviewer 1 Report

Thanks to the authors for revising the manuscript accordingly. But still spelling mistakes are exist in the manuscript. For example Line-261, often due to weight gain and obesity.